# Usefulness of Fecal Calprotectin in the Management of Patients with Toxigenic *Clostridioides difficile*

**DOI:** 10.3390/jcm10081627

**Published:** 2021-04-12

**Authors:** Cecilia Suarez-Carantoña, Argeme Rodriguez-Torres, Adrian Viteri-Noel, Vicente Pintado, Sergio Garcia-Fernandez, Daniel Mora-Pimentel, Rosa Escudero-Sanchez, Fuencisla Martin-Jusdado, Santiago Moreno, Javier Cobo

**Affiliations:** 1Internal Medicine Department, Hospital Ramón y Cajal, CTRA M-607 Colmenar Viejo, Km 9.10, 28034 Madrid, Spain; adrianviteri92@gmail.com; 2Biochemistry Department, Hospital Ramón y Cajal, CTRA M-607 Colmenar Viejo, Km 9.10, 28034 Madrid, Spain; argemert@hotmail.com (A.R.-T.); dmorapimentel@salud.madrid.org (D.M.-P.); fmartin@salud.madrid.org (F.M.-J.); 3Infectious Diseases Department, Hospital Ramón y Cajal, CTRA M-607 Colmenar Viejo, Km 9.10, 28034 Madrid, Spain; vicente.pintado@salud.madrid.org (V.P.); rosa.escudero0@gmail.com (R.E.-S.); smguillen@salud.madrid.org (S.M.); javier.cobo@salud.madrid.org (J.C.); 4IRYCIS (Instituto Ramón y Cajal Investigación Sanitaria), CTRA M-607 Colmenar Viejo, Km 9.10, 28034 Madrid, Spain; 5Spanish Network for Research in Infectious Diseases (REIPI RD16/0016/0011), Instituto de Salud Carlos III, 28034 Madrid, Spain; 6Microbiology Department, Hospital Ramón y Cajal, CTRA M-607 Colmenar Viejo, Km 9.10, 28034 Madrid, Spain; segarciafe@hotmail.com

**Keywords:** *Clostridioides difficile*, fecal calprotectin, *Clostridioides* infection

## Abstract

The availability of highly sensitive molecular tests for the detection of *Clostridioides difficile* in feces leads to overtreatment of patients who are probably only colonized. In this prospective study, the usefulness of fecal calprotectin (fCP) is evaluated in a cohort of patients with detection of toxigenic *C. difficile* in feces. Patients were classified by an infectious diseases consultant blinded to fCP results into three groups—group I, presumed *Clostridioides difficile* infection (CDI); group II, doubtful but treated CDI; and group III, presumed *C. difficile* colonization or self-limited CDI not needing treatment. One hundred and thirty-four patients were included. The median fCP concentrations were 410 (138–815) μg/g in group I, 188 (57–524) μg/g in group II, and 51 (26–97) μg/g in group III (26 cases); *p* < 0.05 for all comparisons. In forty-five out of 134 cases (33.5%), the fCP concentrations were below 100 µg/g. In conclusion, fCP is low in most patients who do not need treatment against *C. difficile*, and should be investigated as a potentially useful test in the management of patients with detected toxigenic *C. difficile*.

## 1. Introduction

*Clostridioides difficile* is the leading cause of nosocomial infectious diarrhea and is one of the most prevalent nosocomial pathogens [1]. Its incidence has increased in recent decades, and it is associated with a significant impact on health and costs [2]. Currently, establishing whether a patient has *C. difficile* infection (CDI) and needs antibiotic therapy is highly important. The administration of oral vancomycin is associated with profound changes in the microbiota that lead to an increased risk of colonization by enterococci, multi-resistant gram-negative bacilli and *Candida* spp. [3,4,5]. Moreover, a recent trial has shown that the treatment of colonized patients with vancomycin does not clear *C. difficile* [6]. In addition, new, high-cost treatments such as fidaxomicin and bezlotoxumab are indicated in patients with a high risk of recurrence [7,8]. Finally, in cases of multiple recurrence, the treatment of choice is fecal microbiota transplantation, a treatment that should not be prescribed lightly [9].

As occurs in most infectious diseases, adequate treatment relies on a quick and efficient diagnosis. Cytotoxicity assays and toxigenic cultures are impractical for the routine management of patients due to their slowness and complexity. Therefore, rapid tests such as toxin detection by means of an enzyme immunoassay (EIA) have been developed, but they lack sufficient sensitivity [10]. This has led to the implementation of nucleic acid amplification tests (NAATs), which are extremely sensitive [11]. Nevertheless, the use of NAATs has led to overdiagnosis, and considerable debate exists regarding how to interpret and manage toxin-negative/NAAT-positive patients [12]. Several authors have shown that a substantial proportion of these patients progress well without treatment against *C. difficile*, emphasizing that NAATs cannot distinguish mere colonization from disease [13,14]. Although the decision to treat should be a clinical one, following the detection of toxigenic *C. difficile* [15] in real-world practice, most patients in whom toxigenic *C. difficile* is detected receive treatment [16,17].

Having an additional tool alongside microbiological tests could avoid misclassifying cases of *C. difficile* colonization as CDI cases. Calprotectin, a 36.5-kDa molecule derived from the cytoplasm of neutrophils [18], is widely used by gastroenterologists in the diagnosis and monitoring of inflammatory bowel disease [19]. Published works have shown that fecal calprotectin (fCP) and lactoferrin levels are higher in patients with CDI than in patients without CDI [20,21]. Additionally, it has been demonstrated that fCP concentrations are higher in severe CDI cases than in mild CDI cases [22,23]. The objective of this investigation was to analyze whether fCP could help in differentiating patients with a positive test for toxigenic *C. difficile* who need antibiotic treatment from those who do not.

## 2. Methods

This was a prospective study carried out at the Ramón y Cajal University Hospital, in Madrid, Spain, from January to December, 2019. All patients diagnosed with toxigenic *C. difficile* in feces were considered for the study, except those diagnosed with pathologies that increase fCP, such as inflammatory bowel disease, ischemic colitis, or colitis due to other enteropathogens.

In our institution, on a regular basis, all positive tests for toxigenic *C. difficile* are regularly reported to one of the infectious disease (ID) consultants in addition to the requesting physician. The ID consultant evaluates each case and makes recommendations on patient management in the first 24 h after diagnosis. For the purposes of this study, cases were classified into three groups: Group I included patients with a highly probable CDI to whom therapy was recommended (presumed CDI) and prescribed; group II (doubtful CDI), when the ID consultant had doubts about the indication of treatment but decided to treat regardless (for example, because of severe immunosuppression or difficulty in distinguishing whether diarrhea could be attributed to other causes) or when the ID consultant recommended not to treat, but the responsible physician decided to treat; and group III (presumed *C. difficile* colonization or self-limited CDI), when the ID consultant recommended not to treat and the patient did not receive treatment against *C. difficile*. Group III patients were only considered to be definitively colonized or as having a self-limited CDI in the event that they did not require treatment for CDI in the month following toxigenic *C. difficile* detection. The ID consultant was blinded to the fCP results when making the recommendations and classifying the episodes. All patients were followed for at least one month.

### 2.1. Laboratory Methods

A 3-step diagnostic algorithm was applied for the detection of toxigenic *C. difficile* in fecal samples based first on the detection of glutamate dehydrogenase (GDH) using an EIA (C Diff Quik Chek, Techlab, Blacksburg, VA, USA), second on toxin A/B detection (TOX A/B Quik Chek, Techlab, Blacksburg, VA, USA), and third, in discordant cases, on PCR amplification for the tcdB gene (BD MAX Cdiff assay, BD Diagnostic, Franklin Lakes, NJ, USA).

Fecal samples in which toxigenic *C. difficile* were detected were frozen and stored. The technique used to determine the fCP level was EIA, which detects protein contained in feces (Calprotectina Blister, Vircell lab, Granada, Spain).

### 2.2. Statistical Analysis

Quantitative variables are presented using absolute and relative frequencies, and continuous variables are presented using the means and standard deviations or medians and quartiles. After checking that values were not normally distributed, fCP was tested among groups I, II, and III using the Kruskal–Wallis test. For post-hoc analysis, the Mann–Whitney U test was used for comparing two groups. Sensitivity, specificity, likelihood ratios and ROC curves were calculated, considering group I as cases of CDI and group III as cases without CDI (not needing treatment). All comparisons were two-tailed. We considered *p*-values < 0.05 to be statistically significant. IBM SPSS software (version 22, IBM Corp, Armonk, NY, USA) was used for statistical analysis.

Written informed consent was obtained from all patients before testing calprotectin in stool samples and for medical chart reviews. The study was approved by the local ethical committee for clinical investigations.

## 3. Results

Two hundred and fifty-one patients were diagnosed with toxigenic *C. difficile* in their feces during the study period. Among them, 134 (53%) patients were included in the analysis after the exclusion of 117 patients for a variety of reasons (Figure 1). The main characteristics of the included patients are shown in Table 1.

Eighty-three patients (62%) were included in group I, 25 (19%) in group II, and 26 (19%) in group III. None of the patients included in group III had to be treated with anti-*C. difficile* drugs in the 30 days following the diagnosis. The median fCP concentrations were 410 (138–815) μg/g in group I, 188 (57–524) μg/g in group II, and 51 (26–97) μg/g in group III. We observed statistically significant differences in the fCP concentration among the three groups (Figure 2).

With a cutoff of 100 μg/g, the sensitivity of fCP was 84.3%, the specificity was 76.9%, and the positive and negative likelihood ratios were 3.65 and 0.20, respectively. The ROC curve value was 0.884. It total, 45 out of 134 (33.5%) cases presented fCP levels below 100 μg/μg. The percentages of cases showing fCP levels below 100 μg/μg in groups I, II, and III were 16.9%, 44%, and 76.9% respectively. fCP levels were also significantly higher in toxin-positive cases than in toxin-negative/NAAT-positive cases (Figure 3).

## 4. Discussion

Our results show that fCP levels were significantly lower in patients who were managed without specific treatment against *C. difficile*, and suggest that fCP could be a useful test to manage patients with toxigenic *C. difficile* in order to avoid unnecessary treatments.

It has been suggested that diagnostic tests could distinguish sick patients from merely colonized or “excretory” *C. difficile* [13]. However, this approach perhaps oversimplifies the clinical reality. In Polage’s study, 40.6% of toxin-negative patients finally received treatment (despite their physicians not having the NAAT results), and another 8% had a subsequent positive toxin test [14]. Moreover, although higher recurrence and severity rates have been found in toxin-positive cases than in toxin-negative/NAAT-positive cases, no higher mortality or complications have been found in a large recent study [17]. As established by the European guidelines for patients with evidence of *C. difficile* but with negative toxin tests, a clinical evaluation is needed because these patients can either have a CDI with undetectable toxin levels or false-negative toxin results or can be potential carriers of toxigenic *C. difficile* [24]. Not even a positive EIA would always force the instigation of treatment. In our cohort, for example, six of the 26 untreated patients had positive toxin results, and their clinical course was favorable without treatment.

The ID consultant was able to manage 26 of the 134 patients (19.4%) without antibiotic treatment against *C. difficile*, and none of the patients had an unfavorable course or subsequently required treatment. However, in the real world, most patients are probably not managed by an expert. On the other hand, it cannot be assumed that there was a margin to have treated fewer patients, since in another 18.6% (group II), the ID consultant had doubts or his recommendations were not followed. Furthermore, the finding of 33.5% of patients with an fCP level below 100 μg/g suggests that a considerable proportion of patients in group II and some in group I could have been managed without antibiotic treatment against *C. difficile*.

We chose a cutoff point of 100 μg/g because it is frequently used in the management of inflammatory bowel disease [19,25] and because it allows us to establish a compromise between the sensitivity and the specificity of the test. However, like any quantitative test, levels close to this cutoff point should be interpreted differently from very distant values. In this sense, it is interesting to note that five of the six untreated patients that showed an fCP concentration above 100 μg/g had values below 300 μg/g. On the other hand, it is possible that some of the untreated patients suffered from mild, self-limiting infections that were resolved with the mere suspension of antibiotic treatment and that, for this reason, the levels of fCP were moderately elevated.

Recently, Kelly et al. failed to determine whether fCP could be used to distinguish colonized patients from CDI patients [26]. However, the design of their study was different from ours since, in their study, the patients labeled “colonized” did not present with diarrhea, and the patients with CDI were all diagnosed by means of a NAAT, so they could have included patients with other causes of diarrhea as CDI patients.

The idea that fCP can help to distinguish patients who need specific treatment against *C. difficile* is biologically plausible since fCP is a marker of intestinal inflammation. In fact, several studies have reported much higher fCP concentrations in patients with severe CDI than in patients with mild CDI [22,23,27,28]. In addition, Barbut et al. found a higher concentration of fCP in patients with a toxin detectable by means of a cytotoxicity assay than in those with no free toxin, suggesting the possible role of fCP in the treatment decision [29]. Our results corroborate this finding, since the fCP concentrations were higher in toxin-positive patients than in toxin-negative/NAAT-positive patients (Figure 3).

Our investigation differs from other previous studies because we did not compare the levels of fCP in relation to other diagnostic tests, but in relation to the indication of treatment. However, we do not think that fCP levels themselves determine the indication for treatment, but rather, that it can be a useful test to help clinicians make decisions. Thus, for example, some clinical trials have shown that the availability of serum procalcitonin levels reduces the use of antibiotics in certain contexts [30]. However, all proclacitonin-based algorithms state that the treatment decision must be made after a complete clinical evaluation.

We must consider certain limitations to our research. The decision to assume the treated patients as CDI cases and the untreated (with a favorable clinical course) as non-CDI cases can be questioned, since there could be treated patients that would have done well without treatment. For this reason, the accuracy of fCP as a diagnostic test for CDI should only be considered tentative.

Second, a systematic study of enteropathogens was not carried out in all samples; therefore, it is possible that in some patients, the concentrations of fCP were elevated due to the presence of other pathogens. Additionally, a significant proportion of patients did not participate in the study for various reasons, and we cannot rule out that this led to biases in the study population. This limitation could have affected the number of patients in the three groups but probably not the fact that most of the untreated patients showed low concentrations of fCP. Finally, in the calculation of the diagnostic accuracy of fCP, we did not include those patients for whom a treatment was prescribed despite the ID consultant advising against it, as well as those for whom he had doubts about the treatment indication. These patients compose a heterogeneous group in which clinical decisions are more difficult and should be the subject of prospective studies to validate the usefulness of fCP concentration in their treatment.

## 5. Conclusions

In summary, our study shows that approximately one-third of patients with detected toxigenic *C. difficile* have low concentrations of fCP, and that fCP concentrations are significantly lower in patients who are managed without a specific treatment against *C. difficile*. Future clinical trials or intervention studies should evaluate the effects of the simultaneous availability of fCP and microbiological tests on reducing unnecessary treatments against *C. difficile*.

## Figures and Tables

**Figure 1 jcm-10-01627-f001:**
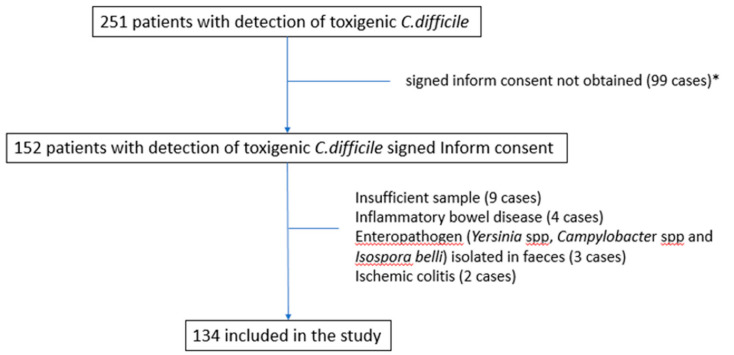
Flow-chart of patients included in the cohort. * Informed consent was not obtained for several reasons (dementia, patient not admitted to the hospital, non-availability of investigation team, short life expectancy, case not attended to by the infectious diseases team, etc.).

**Figure 2 jcm-10-01627-f002:**
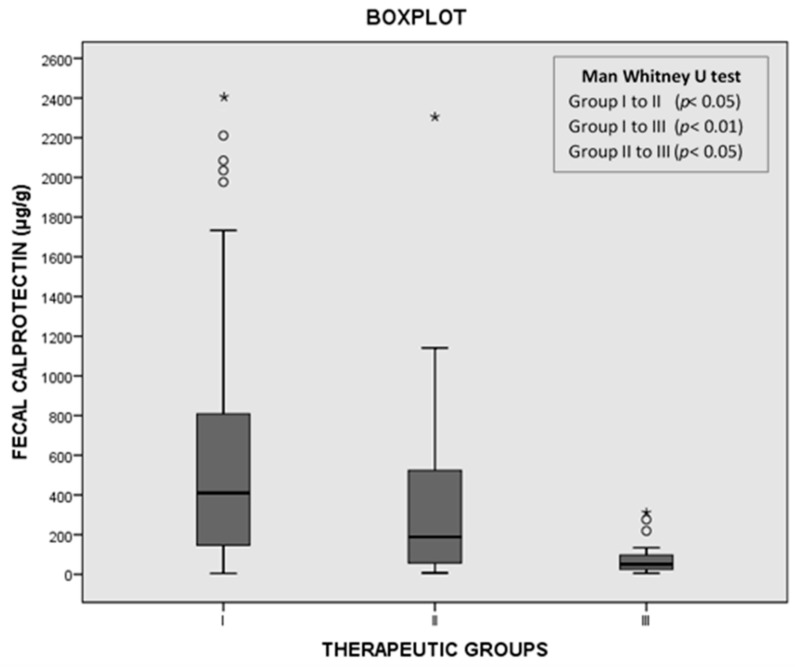
Distribution of fecal calprotectin (fCP) levels among groups I, II, and III. * maximum value in the data, ° potential outlier.

**Figure 3 jcm-10-01627-f003:**
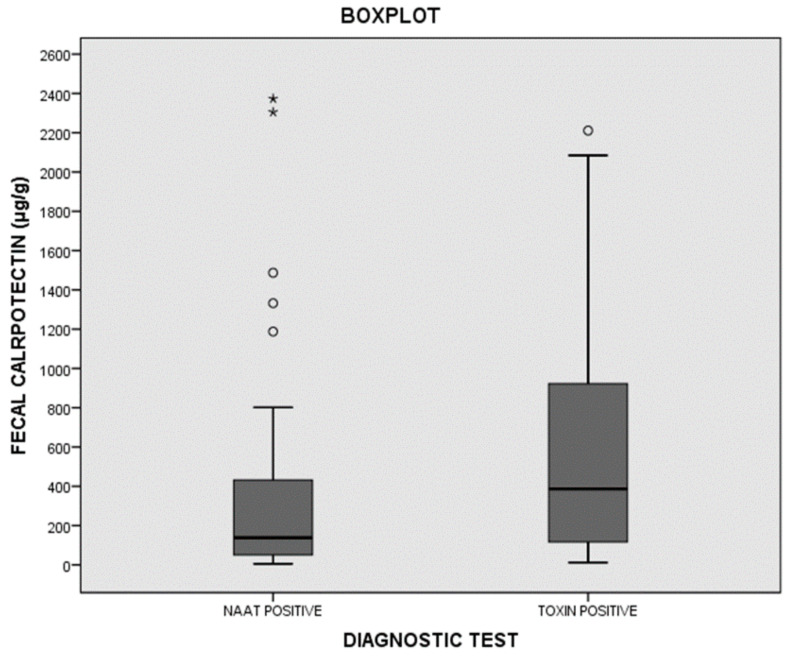
Distribution of fecal calprotectin (fCP) among toxin-positive and toxin-negative /NAAT-positive cases. * maximum value in the data, ° potential outlier.

**Table 1 jcm-10-01627-t001:** Comparison of cohort patients by groups (presumed *C. difficile* infection treated, group I; doubtful *C. difficile* infection treated, group II; and non-treated patients, group III).

Characteristic		All Patients *n* = 134	Group I *n* = 83	Group II *n* = 25	Group III *n* = 26	*p*-Value
Gender (male)		68 (50.7)	42 (50.6)	9 (36.0)	17 (65.4)	0.111
Age (years)		73 (60–83)	78 (64–84)	71 (60–79)	67 (50–79)	0.059
Diabetes		44 (30.6)	30 (36.1)	3 (12.0)	8 (30.8)	0.072
CKD (≥stage III)		31 (23.1)	21 (25.3)	3 (12.0)	7 (26.9)	0.882
Solid tumor		35 (26.1)	15 (18.1)	13 (52.0)	7 (26.9)	0.017
Hematologic malignancy		10 (7.4)	6 (7.2)	2 (8.0)	2 (7.7)	0.274
Solid organ transplantation		15 (11.2)	8 (9.6)	5 (20.0)	2 (7.7)	0.357
Moderate or severe dependence *		19 (14.1)	12 (14.4)	2 (8.0)	5 (19.2)	0.616
Previous CDI		31 (23.1)	20 (24.1)	4 (16.0)	7 (26.9)	0.616
Antibiotic use during last month		108 (82.1)	69 (83.1)	21 (84.0)	18 (69.2)	0.499
No. of stools/day		5 (3–6)	6 (4–7)	5 (3–6)	3 (2–5)	0.000
Time to resolution of diarrhea		3 (2–5)	4 (2–5)	3 (2–3)	1 (1–2)	0.043
Toxin positive		61 (45.5)	44 (53.1)	11 (44.0)	6 (23.1)	0.026
Toxin-negative/NAAT-positive		73 (54.4)	39 (46.9)	14 (53.8)	20 (76.9)	0.665
Fecal calprotectin (μg/g)		257 (67–592)	410 (138–815)	188 (57–524)	51 (26–97)	0.000
Severity &	Non-severe	81 (75.0)	60 (72.3)	21 (84.0)	NA	
	Severe	25 (23.1)	21 (25.3)	4 (16.0)	NA	
	Fulminant	2 (1.8)	2 (2.4)	0	NA	
Fever		44 (32.8)	32 (38.5)	8 (32.5)	4 (15.4)	0.089
Leukocyte count		9865 (6735–14,900)	11850 (7420–17,100)	8510 (5900–10,700)	8620 (6224–10,000)	0.026
C-reactive protein (mg/L)		67 (29–127)	66 (28–126)	105 (52–149)	41 (14–104)	0.149
Creatinine (mg/dL)		0.9 (0.6–1.7)	0.9 (0.6–1.9)	0.8 (0.6–1.0)	0.9 (0.7–1.8)	0.207
Albumin (mg/dL)		2.5 (2.0–3.0)	2.5 (2.0–2.9)	2.3 (2.0–2.9)	2.8 (2.2–3.1)	0.234
Recurrence (8 weeks)		15 (13.8)	14 (16.0)	1 (4.0)	NA	0.185
Death (all causes) at 30 days		11 (8.2)	7 (8.4)	2 (8.0)	2 (7.7)	0.333

& according to IDSA guidelines. * Barthel index score > 55. CKD, chronic kidney disease; CDI, *C. difficile* infection; NAAT, nucleic acid amplification test.

## Data Availability

The data presented in this study are available on request from the corresponding author. The data are not publicly available the data is not publicly available because it contains information that could compromise the privacy of research participants.

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
