# Peer review of "Usefulness of Fecal Calprotectin in the Management of Patients with Toxigenic Clostridioides difficile"

_jcm, 2021, doi:10.3390/jcm10081627_

Round 1

Reviewer 1 Report

Thank you for the opportunity to review this manuscript. Cecilia Suarez-Carantoña and colleagues have conducted an interesting study looking at whether faecal calprotectin (fCP) correlates with the decision to treat Clostridioides difficile infection by an Infectious Diseases physician. This is a novel take on the potential role of fCP in the management of C. difficile infection (CDI). Overall the research design was appropriate and manuscript is well written, with no significant issues. It should be of interest to a wide range of clinicians and will hopefully prompt further research in this area. The authors may wish to consider the following, mostly minor,  modifications to their manuscript.

Methods:

Can the authors clarify whether the same ID physicians reviewed all of the cases, or was it one of a team (and how many were there in the team)? Did the authors exclude all patients with other causes of a raised fCP? Figure 1 suggests that they did, but it would be good to clarify this in the text. Did they, for example, consider non-steroidal anti-inflammatory use?

Results:

Almost half of the cases were excluded (117/251, 47%). The authors rightly comment on this limitation in the discussion. Did they consider ways of increasing participation (e.g. seeking consultee approval when potential participants lacked capacity to provide informed consent)?

Rather than listing only certain comorbidities in Table 1 could the authors include a composite measure of comorbidity, such e.g. Charlson index? Could they consider including more clinical and biochemical parameters, e.g. body temperature, change in serum creatinine? The three groups appear to differ in some respects (e.g. gender, age, leucocytosis) but not others. Can the authors apply a statistical analysis to each of the factors they have listed across the three groups?

Did the authors look at the performance of fCP at other cut off limits, or just choose 100 ug/g as it is used for the diagnosis of IBD?

Author Response

Thank you very much for your review and recommendations. 

METHODS:

Can the authors clarify whether the same ID physicians reviewed all of the cases, or was it one of a team (and how many were there in the team)?

In our hospital, an Infectious diseases consultant (JC) reviews and provides advice on most cases of C difficile Infection. However, when the patient is already being cared for by another infectious disease consultant, this is usually the one who makes the recommendation. We have not exactly quantified it but probably 70-80% of the recommendations were made by the same consultant (JC).

Did the authors exclude all patients with other causes of a raised fCP? Figure 1 suggests that they did, but it would be good to clarify this in the text. Did they, for example, consider non-steroidal anti-inflammatory use

Yes. We exclude patients diagnosed with other conditions that clearly produce marked increases in fCP, such as other intestinal infections, ischemic colitis and, above all, inflammatory bowel diseases. Patients receiving non-steroidal anti-inflammatory drugs were not excluded because, as far as we know, the increases in fCP that they can produce are scant (Rendek, 2016).

According to the suggestion of the reviewer we have added the following sentence to the methods section. “This is a prospective study carried out at the Ramón y Cajal University Hospital, in Madrid, Spain, from January to December 2019. All patients diagnosed with toxigenic C. difficile in feces were considered for the study, except those diagnosed with pathologies that increase fCP, such as inflammatory bowel disease, ischemic colitis, or colitis due to other enteropathogens”.

RESULTS:

Almost half of the cases were excluded (117/251, 47%). The authors rightly comment on this limitation in the discussion. Did they consider ways of increasing participation (e.g. seeking consultee approval when potential participants lacked capacity to provide informed consent)?

Thanks for the comment. The greatest difficulty in recruitment was due to the impossibility of obtaining informed consent, especially in three situations:

- patients discharged from the emergency room

- patients diagnosed in primary care

- patients with cognitive deficits without the ability to understand the nature of the study

We must recognize that we did not establish special measures to obtain informed consent in these patients because the incidence of C.difficile infection is quite high in our hospital, and we had no problems recruiting enough cases. We do not consider that these biases would be very relevant, but we agree that they may represent a limitation of our study that can be improved in future investigations. However, the fact that the recurrence rate (around 14%) and the proportion of toxin + cases (47%) fully coincide with our global experience in recent years, makes us think that the population studied is representative.

Rather than listing only certain comorbidities in Table 1 could the authors include a composite measure of comorbidity, such e.g. Charlson index? Could they consider including more clinical and biochemical parameters, e.g. body temperature, change in serum creatinine? The three groups appear to differ in some respects (e.g. gender, age, leucocytosis) but not others. Can the authors apply a statistical analysis to each of the factors they have listed across the three groups?

Following the reviewer's recommendation, we have added some variables (dependence measured by Barthel index score, PCR, albumin concentration and creatinine concentration), as well as the level of statistical significance (p-value) in table 1. Please find attached table 1 in a word document.

Did the authors look at the performance of fCP at other cut off limits, or just choose 100 ug/g as it is used for the diagnosis of IBD?

Yes. We considered other cut-off points with the statistician, the cut-off point of 100 ug/g worked better than other cut offs.

With a cut off of 50 μg/g:  sensitivity  92.8%; specificity 50.0%

With a cut off of 100 μg/g: sensitivity 84.3%; specificity 76.9%

With a cut off of 200 μg/g, sensitivity of fCP was 71.1%, specificity 88.5%

The cut off on 100 ug/g had the added advantage that it is the level that is frequently used for the monitoring of inflammatory bowel disease. In any case, our objective with this study was not to find cut-off points, but to show the "proof of concept" that fCP is low in patients with detection of toxigenic C difficile who are not treated and have a good outcome.

Reviewer 2 Report

I add a file with comments and suggestions

Author Response

Thank you very much for the suggestions. We have integrated some of the references provided by the reviewer.

Reviewer 3 Report

I appreciate that I had a chance to review this manuscript.

This manuscript contain significant findings about management of patients with toxigenic Clostridioides difficile.

I had a few comments for this manuscript.

  1. Could you describe the criteria of Group â… , Group â…¡ and Group â…¢ more clearly? It seems that the patients were divided according to the Infectious diseases (ID) consultants’ subjective judgment rather than the objective criteria.
  2. It would be said that fCP was significantly higher in the patients who were treated as CDI, but it would not be said that fCP could be used as a biomarker to decide the treatment of CDI. If the authors wish to conclude that fCP could be used as a biomarker of CDI, I recommend to divide the patients according to the level of fCP and compare the severity of CDI.
  3. The more detail explanations of Figure 3 would be needed. What do the authors mean by Figure 3?
  4. The discussion should be written along with the results. I could not find the explanation and discussion why fCP was higher in Group â…  (CDI treated) patients. It would be better to describe the meanings why fCP was elevated in the patients who were treated as CDI.  

Author Response

Thank you very much for your review and recommendations.

  1. Could you describe the criteria of Group â… , Group â…¡ and Group â…¢ more clearly? It seems that the patients were divided according to the Infectious diseases (ID) consultants’ subjective judgment rather than the objective criteria.

 The classification of the groups was objective: I: cases treated as CDI (undoubtedly for the ID consultant); II: cases treated but in which there were doubts for different reasons or the treatment was done despite the opinion of the ID consultant, and III: untreated cases because the patient was considered to be only colonized or the episode was self-limited.

We have modified the wording of the paragraph as follows:

“For purposes of the study cases were classified into three groups: Group I included patients with a highly probable CDI to whom therapy was recommended (presumed CDI) and prescribed; group II (doubtful CDI), when the ID consultant had doubts about the indication of treatment but decided to treat regardless (for example, because of severe immunosuppression or difficulty in distinguishing whether diarrhea could be attributed to other causes) or when the ID consultant recommended not to treat, but the responsible physician decided to treat; and group III (presumed C.difficile colonization or self-limited CDI), when the ID consultant recommended not to treat and the patient did not receive treatment against C.difficile. Group III patients were only considered definitively colonized or as having a self-limited CDI in the event that they did not require treatment for CDI in the month following toxigenic C.difficile detection”

  1. It would be said that fCP was significantly higher in the patients who were treated as CDI, but it would not be said that fCP could be used as a biomarker to decide the treatment of CDI. If the authors wish to conclude that fCP could be used as a biomarker of CDI, I recommend to divide the patients according to the level of fCP and compare the severity of CDI.

As mentioned in the manuscript, it has already been previously shown that fCP is higher in severe than in mild CDI. In our study, this can also be glimpsed since in group I there are more severe cases than in group II, and the levels of fCP are also higher in group I than in II.

What our research shows is that there is a remarkable proportion of cases with positive tests for toxigenic C. difficile that have low levels of fCP, and that many untreated patients -by clinical decision, without access to fCP- with good outcomes, have low fCP, which suggests that fCP could be used as a test to aid the decision not to treat in case of doubt.

  1. The more detail explanations of Figure 3 would be needed. What do the authors mean by Figure 3?

In Figure 3 we confirm the findings already described previously by Barburt et al. The fCP levels are higher in toxin + cases than in toxin- / CRP + cases.

It is generally accepted that there is more "burden of disease" in toxin + cases than in toxin- / NAAT + cases. However, it is not appropriate to make treatment decisions solely by the pattern of diagnosis. In other words, there are cases of toxin- / NAAT + patients who have obvious C.difficile infection and need treatment.

In the discussion we have added a reference to this figure.

  1. The discussion should be written along with the results. I could not find the explanation and discussion why fCP was higher in Group â…  (CDI treated) patients. It would be better to describe the meanings why fCP was elevated in the patients who were treated as CDI.  

We believe that, throughout the introduction and discussion of our manuscript, the idea can be followed that a severe C. difficile infection will generally have high levels of fCP and a mild one will have lower levels. What we suggest in this research is that if a clinician has doubts to whether a patient with positive tests for toxigenic C.difficile is really suffering from C.difficile diarrhea or, perhaps, he/she is only colonized and has diarrhea from another cause (drugs, enteral nutrition, irritable bowel syndrome ...) fCP levels can help to make the decision.

Round 2

Reviewer 3 Report

I confirmed the revision of manuscript.